# *Ganoderma lucidum* Immobilized on Wood Demonstrates High Persistence During the Removal of OPFRs in a Trickle-Bed Bioreactor

**DOI:** 10.3390/jof11020085

**Published:** 2025-01-22

**Authors:** Shamim Tayar, Javier Villagra, Núria Gaju, Maira Martínez-Alonso, Eduardo Beltrán-Flores, Montserrat Sarrà

**Affiliations:** 1Departament d’Enginyeria Química Biològica i Ambiental, Escola d’Enginyeria, Universitat Autònoma de Barcelona, 08193 Bellaterra, Barcelona, Spain; shamim.tayar@uab.cat (S.T.); eduardo.beltran@uab.cat (E.B.-F.); 2Departament de Genètica i Microbiologia, Universitat Autònoma de Barcelona, 08193 Bellaterra, Barcelona, Spain; javier.villagra@uab.cat (J.V.); nuria.gaju@uab.cat (N.G.); maira.martinez@uab.cat (M.M.-A.)

**Keywords:** flame retardants, bioreactor, immobilization, white-rot fungi, microbial assessment

## Abstract

Emerging pollutants such as organophosphate flame retardants (OPFRs) pose a critical threat to environmental and human health, while conventional wastewater treatments often fail to remove them. This study addresses this issue by evaluating the bioremediation potential of white-rot fungi for the removal of two OPFRs: tris(2-chloroethyl) phosphate (TCEP) and tributyl phosphate (TBP). Three fungal species—*Ganoderma lucidum*, *Trametes versicolor*, and *Phanerochaete velutina*—were screened for their degradation capabilities. Among these, *G. lucidum* and *T. versicolor* demonstrated removal efficiencies exceeding 99% for TBP, while removal rates for TCEP were significantly lower, with a maximum of 30%. The exploration of the enzyme role showed that cytochrome P450 is involved in the degradation while the extracellular laccase is not involved. Continuous batch experiments were performed using a trickle-bed reactor (TBR) operating under non-sterile conditions, a setting that closely resembles real-world wastewater treatment environments. *G. lucidum* was immobilized on oak wood chips, and the removal efficiencies were measured to be 85.3% and 54.8% for TBP and TCEP, respectively, over 10 cycles. Microbial community analysis showed that *G. lucidum* remained the dominant species in the reactor. These findings demonstrate the efficacy of fungal-based trickle-bed bioreactors, offering a sustainable and efficient alternative for addressing environmental pollution caused by highly recalcitrant pollutants.

## 1. Introduction

Flame retardants are synthetic additives that have been widely used since the 1960s to improve fire safety by inhibiting combustion and slowing the spread of fire in various consumer products, such as electronic devices, building materials, fabrics, and furniture [1,2]. Due to regulatory bans on the use of polychlorinated biphenyls and polybrominated diphenyl ethers in the European Union and their voluntary phase-out in the United States, the use of phosphorus-based flame retardants (PFRs) has increased as a potential alternative to brominated flame retardants [3]. Flame retardants are typically classified in two ways. The first classification is based on their chemical composition, distinguishing between inorganic, organic (including phosphinates, phosphonates, and organophosphate esters (OPEs), commonly referred to as organophosphate flame retardants (OPFRs)), and halogenated compounds [3]. The second classification is based on how they are incorporated into materials, differentiating between reactive and additive types. Reactive PFRs are chemically bonded into the polymer structure, reducing the risk of loss over time, while additive PFRs are physically mixed with the polymer, which may lead to a decrease in concentration over time and potentially compromise the material’s flame retardancy.

OPFRs play two distinct roles: halogenated OPFRs function as flame retardants, while non-halogenated OPFRs are primarily used as plasticizers. Notable examples of these compounds include tributyl phosphate (TBP) and tris(2-chloroethyl) phosphate (TCEP), which are widely applied in products such as floor polish, lacquers, and hydraulic fluids. In the production of polyvinyl chloride (PVC), phosphorus-containing compounds like triethyl phosphate, tris(1-chloro-2-propyl) phosphate, and TCEP are commonly used as plasticizers. Beyond PVC manufacturing, organophosphates are also incorporated into textiles, rubber, electronic devices, and various resins across a broad range of industrial applications [3].

The volatile nature of additive OPFRs allows them to easily evaporate from products into indoor air. In contrast, reactive OPFRs are chemically bonded to polymers, making them more stable and providing a long-lasting plasticizing effect. However, even though reactive OPFRs are non-volatile, they can still migrate from deteriorating materials and microplastics into wastewater treatment plants, rivers, and eventually oceans, raising environmental concerns [4]. The widespread use of OPFRs is anticipated to contribute significantly to environmental pollution. The first report of organophosphate esters (OPEs) in the environment was written by Sheldon and Hites in 1978, who detected TBP, tri(2-butoxyethyl) phosphate, and triphenyl phosphate in freshwater samples [5]. Subsequent research has revealed the extensive presence of OPEs in various environmental media, including indoor air, dust, water, sediments, and biota, with concentrations ranging from parts per trillion (ppt) to hundreds of parts per million (ppm) [6]. The frequent detection and persistence of OPFRs suggest multiple exposure pathways for humans, such as ingestion or inhaling indoor dust and consuming contaminated food or water. Prolonged exposure and accumulation of OPFRs in animals and humans have been linked to adverse health effects, including renal toxicity, neurotoxicity, reproductive dysfunction, carcinogenicity, and endocrine disruption [7,8]. Lipophilic OPFRs are known to bioaccumulate, with evidence of their presence in human hair, nails, urine, and breast milk [9,10]. Chlorinated OPFRs, such as TCEP, tris(1-chloro-2-propyl) phosphate, and tris(1,3-dichloro-2-propyl) phosphate, have been identified as neurotoxic and carcinogenic [2,11]. Additionally, certain OPEs pose specific risks to both animal and human health; for instance, triethyl phosphate has been shown to disrupt embryonic development in chickens, reducing growth and pipping success and lowering plasma thyroxine levels [12]. TBP exposure has also been linked to liver effects in mice, including the development of tumors [13].

Given the harmful effects of OPFRs, it is crucial to develop effective methods for their removal from the environment. However, conventional activated sludge treatment in wastewater treatment plants (WWTPs) has been shown to achieve only a 49% degradation rate for OPFRs, with a significant portion being released into freshwater or marine ecosystems [14]. Alarmingly, chlorinated OPFRs, such as TCEP, present a particular challenge for biodegradation, as traditional activated sludge processes have proven to be largely ineffective in their removal [15]. Fortunately, bioremediation offers a cost-effective, environmentally friendly, and efficient solution for addressing a wide range of pollutants, including OPFRs, compared to conventional physical–chemical methods. For instance, microbial degradation of organophosphorus compounds has been extensively reviewed [16,17]. However, bacterial remediation of emerging contaminants faces two major challenges: (1) bacteria require an adaptation period to degrade certain contaminants effectively, and (2) they struggle to degrade pollutants below a specific concentration threshold, often referred to as the “response threshold” [18,19]. Fungal bioremediation, particularly using white-rot fungi (WRF), can overcome these limitations and has demonstrated success in degrading various micropollutants, including those resistant to bacterial degradation [20,21,22]. The remarkable degradation capability of WRF is attributed to their effective enzymatic systems, which include extracellular ligninolytic enzymes such as laccase, as well as intracellular systems like cytochrome P450 [23,24,25].

TCEP and TBP are bio-recalcitrant organophosphate compounds frequently classified as emerging contaminants due to their association with wastewater pollution in freshwater systems [26,27,28,29,30,31]. These compounds exhibit exceptional stability in the natural environment, showing resistance to conventional degradation processes such as photolysis and hydrolysis [32]. Given the persistence and environmental impact of these pollutants, exploring alternative methods for their effective removal is critical.

This study investigates the ability of WRF (*Ganoderma lucidum* (*G. lucidum*), *Trametes versicolor* (*T. versicolor*), and *Phanerochaete velutina* (*P. velutina*)) to degrade OPFRs, focusing on their capacity to break down TCEP and TBP. A key aspect of this research is the evaluation of *G. lucidum* in a trickle-bed reactor (TBR) operating under non-sterile conditions. This approach tests the fungus’s degradation efficiency and accounts for the environmental complexity encountered in real-world applications, where microbial competition and non-sterile conditions prevail. The use of *G. lucidum* immobilized on wood chips within a continuous TBR setup serves to simulate long-term pollutant exposure, offering insights into the fungus’s persistence and functionality under such conditions. This work highlights the ability of *G. lucidum* to degrade the resistant TCEP, positioning it as a promising candidate for addressing pollutants that challenge conventional treatment methods.

## 2. Materials and Methods

### 2.1. Microorganisms, Media, and Cultivation

*T. versicolor* ATCC 42530 was acquired from the American Type Culture Collection (Manassas, VA, USA), *P. velutina* FBCC941 (T244i) was obtained from the Fungal Biotechnology Culture Collection of the University of Helsinki (FBCC, Helsinki, Finland), and *G. lucidum* FP-58537-Sp was kindly provided by Dr. C.A. Reddy from the Department of Microbiology and Molecular Genetics and the NSF Center for Microbial Ecology, Michigan State University (East Lansing, MI, USA). All fungal species were maintained in subcultures made every 30 days on malt extract agar plates (2% *w*/*v*) at 25 °C. Mycelial suspensions and pellets were prepared according to the previously described method in a malt extract medium containing 20 g/L malt extract (Scharlau, Barcelona, Spain) adjusted to pH 4.5 before sterilization [33]. The defined medium used for batch degradation experiments consisted of glucose (8 g), ammonium tartrate (3.3 g), dimethyl succinate (1.68 g), micronutrient solution (10 mL), and macronutrient solution (100 mL) in 1 L of deionized water [34]. The pH was adjusted to 4.5 by 1M HCl and 1M NaOH solutions. For TBR experiments, 250 mL of mycelial suspension was used to inoculate 1 kg of autoclaved oak wood chips (sieve mesh sizes between 7.10 and 16 mm) according to the reported method [23].

### 2.2. Chemicals and Reagents

Analytical-grade (purity ≥ 98% 99%) TBP (CAS No. 126-73-8) and TCEP 97% (CAS No. 115-96-8), laccase mediator 2,2-azino-bis (3-ethylbenzothriazoline-6-sulphonic acid) diammonium salt (ABTS 98%), 2,6-dymetoxyphenol (DMP, 99%), commercial laccase purified from *T. versicolor* (20 U/mg), sodium malonate dibasic monohydrate, dimethyl succinate, ammonium acetate, and cytochrome P450 (CYP450) inhibitor 1-aminobenzotriazole (ABT) (98% pure) were purchased from Sigma-Aldrich Co. (Barcelona, Spain). The Microtox bioassay kit was supplied by Strategic Diagnostics Inc. (Newark, NJ, USA). Cyclohexane was purchased from Labkem (Barcelona, Spain). Chromatographic-grade methanol, acetonitrile, and formic acid (≥98%) were obtained from Merck (Darmstadt, Germany). All other chemicals used were analytical grade and acquired from Sigma-Aldrich Co. (Barcelona, Spain).

### 2.3. Screening of Fungal Strains for OPFR Removal in Erlenmeyer Flasks

Degradation experiments were conducted in 250 mL Erlenmeyer flasks, each containing 50 mL of defined medium spiked with either TBP or TCEP at a final 10 mg/L concentration. The medium was prepared under sterile conditions using a concentrated stock solution in methanol (MeOH). The flasks were inoculated with pellets of the three WRF species, achieving a fungal biomass concentration of approximately 3.5 g dry weight (DW)/L. The cultures were incubated at 25 °C and subjected to orbital agitation (135 rpm) for 7 days. Two control setups were prepared to assess the contribution of abiotic factors and non-active fungal biomass: an uninoculated abiotic control and a heat-killed control, where fungal cultures were autoclaved at 120 °C for 30 min. Each experiment, including the controls, was performed in triplicate.

### 2.4. Assessment of the Role of Cytochrome P450 in the Degradation of TBP and TCEP by G. lucidum

To investigate the role of CYP450 during TBP and TCEP decomposition by *G. lucidum*, in vivo degradation experiments were performed. Specifically, ABT was fortified into both pellet-TBP and pellet-TCEP incubation systems to reach a final concentration of 5 mM [35]. The initial biomass and pollutant concentrations were set at 3.5 g DW/L and 10 mg/L, respectively. Control setups included an abiotic system and experimental control (inhibitor-free system), and all cultures were incubated at 25 °C under continuous orbital-shaking at 135 rpm for 15 days. Each experimental group consisted of three replicates, with samples collected at specified time intervals to measure TBP and TCEP concentrations.

### 2.5. Screening of Fungal Growth on Oak Wood Chips

The lignocellulosic substrate (oak wood chips) was autoclaved at 120 °C for 30 min, then immersed in tap water and strained to remove excess water under sterile conditions. Cultures were set up in 250 mL borosilicate glass bottles Schott-Duran GLS 80s (95 × 105 mm, Duran Inc., East Haddam, CT, USA) equipped with a single port fitted with a 0.45 μm filter for passive air intake. Under sterile conditions, 10 g of the sterilized substrate, at full water-holding capacity, was placed into each bottle and inoculated with 3 mL of mycelial suspension from the three fungal species. Each experiment was performed in triplicate. The cultures were maintained under static conditions at 25 °C for 30 days. Samples were collected at specified intervals to measure ergosterol content, which served as the basis for biomass quantification using a modified method described previously [36].

### 2.6. TCEP and TBP Removal in a TBR Under Non-Sterile Conditions

The lab-scale trickle-bed reactor (TBR) essentially consists of a vertical cylindrical fixed bed, a liquid recirculation loop, and a pH maintenance system (Appendix A). Wood chips (1 kg) colonized by *G. lucidum* were transferred into a methacrylate tube (Ø 8.5 cm, H 58 cm) and supported by a mesh, achieving an approximate working volume of 2.5 L and a porosity of 60%. The synthetic wastewater (SWW) was prepared with tap water, adding 100 mL of macronutrients, 10 mL of micronutrients, and 10 mg of both OPFRs per liter. The SWW was loaded into the packing bed from the top of the reactor through a rotary distributor and then collected by the reservoir tank placed at the bottom. The collected water was mixed with a magnetic stirrer, and its pH was maintained at 4.5 by adding either 1 M HCl or 1 M NaOH [37]. An external bottom-to-top recirculation loop (200 mL/min) was provided, by which the collected water was continuously fed into the packing bed. Simultaneously, another two identical reactors filled with heat-killed colonized wood (killed control, KC) and non-colonized chips (abiotic control, AC) were operated in parallel to assess the adsorption from the biomass and lignocellulosic supporting material, respectively. Multiple runs were implemented in sequencing batch mode (2-day cycle) at room temperature. Samples were taken from the tank after each batch to measure TBP and TCEP concentration, turbidity, and chemical oxygen demand (COD), and then the SWW was completely replenished.

### 2.7. Contact Time in the TBR

Residence time and the total contact time were measured using a volumetric quantification method as previously described [38]. In brief, oak wood chips (free of fungi) were soaked in water, and the pump was activated for 3.5 min. The first drop at the outlet appeared after 20 s. Subsequently, all the introduced distilled water was collected at the outlet, and the outflow volume was measured at regular intervals. A flow rate of 200 mL/min was used for all experiments, and each test was performed in triplicate.

### 2.8. Analytical Methods

#### 2.8.1. Biomass Quantification in Liquid Cultures

Biomass in the broth was measured as dry weight after filtration with Whatman GF/C glass fiber filters (Whatman, Maidstone, UK) and drying at 105 °C to a constant weight.

#### 2.8.2. Ergosterol Extraction and Quantification

The immobilized biomass on the lignocellulosic substrate was quantified by adopting a modified method documented elsewhere [7]. Briefly, after triturating by an analytical mill (A 11 basic, IKA GmbH, Staufen, Germany), 0.5 g of each homogeneously mixed sample was transferred into a test tube, together with 1 mL of cyclohexane and 3 mL of a KOH-methanol solution (10%, *w*/*v*). The mixture was subsequently sent for ultrasonication for 15 min (50/60 Hz, 360 W), which was followed by heating at 70 °C for 90 min. Then, 1 mL of distilled water and 2 mL of cyclohexane were added before vortexing for 30 s and centrifuging at 3500 rpm for 5 min. The upper organic phase was collected, while the aqueous phase was washed twice with 2 mL of cyclohexane. The organic phase was evaporated to dryness by N_2_ under a moderate stream. The residue was re-dissolved in 1 mL of methanol at 40 °C for 15 min. Afterward, it was vortexed for 30 s and centrifuged at 6000 rpm for 3 min. The resultant solution was kept in a 2 mL amber vial at −20 °C prior to analysis. The quantification was achieved through an HPLC (Ultimate 3000, Dionex, Sunnyvale, CA, USA) equipped with a UV detector at 282 nm, and using a C18 reverse-phase column (Phenomenex^®^, Kinetex^®^ EVO C18 100 A, 4.6 mm × 150 mm, 5 µm). The isocratic elution was performed using acetonitrile (100%) at 1 mL/min with an oven temperature of 35 °C, under which the retention time was 7.593 min. The injection volume was 40 µL [38].

#### 2.8.3. OPFR Concentration

The liquid chromatography studies were performed on a Prominence UFLC that was controlled by an LCMS solution Chromatography Data System software (version 3), consisting of a SIL/20A autosampler and LC-20AD solvent-delivery system, and equipped with an LCMS-2010A detector, all of which were from Shimadzu (Kyoto, Japan). The separations were carried out on a Purospher^®^ Star RP-18 end-capped (5 µm) column (Merck KGaA, Darmstadt, Germany) at a 30 °C column oven temperature with ultrapure water (A) and acetonitrile (B), which was pumped at 0.25 mL/min in gradient mode (total run time: 36 min). The percentage of (B) was changed as follows: 0 min, 50%; 4 min, 50%; 5 min, 80%; 7 min, 80%; 8 min, 90%; 13 min, 90%; 21 min, 100%; 26 min, 100%; 35 min, 50%; and 36 min, 50%. The sample injection volume was 2 µL. MS analysis was conducted under selected-ion-monitoring (SIM) positive-mode electrospray ionization (ESI+) with the mass of *m*/*z* 267.10 Da and 286.90 Da. Data in the tuning file were selected for parameters such as lens voltage values, interface, Q-array, and others. High-purity nitrogen was used as the nebulizing gas at 1.50 L/min, and the drying gas pressure was 0. The other interface parameters were optimized as follows: the interface, CDL, and heat block temperatures were 250 °C, and the detector voltage was set at 1.5 kV.

#### 2.8.4. TBP and TCEP Extraction from Fungi

Samples underwent freeze-drying and extraction using a 15 mL hexane/acetone (1:1 *v*/*v*) mixture. An ultrasound-assisted extraction (UAE) was performed for 15 min, followed by centrifugation at 3220 r.c.f. for 5 min. Both steps were repeated twice, and the resulting extracts were collected and evaporated to dryness at 20 °C under a nitrogen stream. Subsequently, 5 mL of hexane/methanol (1:3) was added, which was followed by an additional centrifugation step at 3220 r.c.f. for 5 min. An aliquot of 200 µL was then collected and spiked with 15 ng of the IS mixture.

#### 2.8.5. TBP and TCEP Extraction from Wood

Approximately 1 kg of dried wood was exposed to a 10 mg/L solution of TNBP and TCEP. This procedure was repeated 10 times. Samples were sieved to 125 µm prior to extraction. Then, 1 g was weighted and mixed with 0.5 g of copper and 100 ng of IS, and an accelerated solvent extraction (ASE) was performed with hexane/acetone (1:1 *v*/*v*). Extracts were evaporated to dryness under a nitrogen stream and reconstituted with 500 µL of methanol.

#### 2.8.6. Identification of Transformation Products

Degradation experiments were performed as described in Section 2.3. After 7 days, the content of each Erlenmeyer flask was filtrated with a WhatmanTM glass microfiber filter (GF/A, 47 mm) (Cytiva, Marlborough, MA, USA) to separate the biomass. Then, 20 mL of the filtrate was mixed with 50 µL of internal standard (IS), which was followed by adding filtrate up to 50 mL. Afterward, both liquid and solid samples were kept at −20 °C prior to analysis. Identification of TPs was performed on an ultra-high-performance liquid chromatograph coupled to a hybrid quadrupole-Orbitrap mass spectrometer (UHPLC-Q-Exactive, Thermo Fisher Scientific, Waltham, MA, USA). Chromatographic separation was carried out on a Purospher STAR RP-18 end-capped (2 μm) Hibar HR 150–2.1 mm column (Merck, Darmstadt, Germany). The mobile phase used was (A) acetonitrile and (B) water containing formic acid 0.1% (*v*/*v*) and 5 mM ammonium acetate at 0.2 mL/min in gradient mode. The percentage of (A) was changed as follows: 0 min, 20%; 1 min, 20%; 8 min, 95%; 13 min, 95%; 13.5 min, 20%; and 15 min, 20%. The sample injection volume was 10 μL.

Related to the mass spectrometric conditions, an electrospray ionization (ESI) source was used working in the positive mode under a capillary voltage of 3000 V. Capillary and probe heater temperatures were set at 350 °C and 300 °C, respectively, while sheath and auxiliary gas flow rates were set at 40 and 10 arbitrary units, respectively.

Spectra were acquired in two consecutive scans. First, a full scan in the range of 50–700 *m*/*z* was performed at a resolution of 70,000. Then, an MS/MS scan was carried out for all the compounds that had arrived in the analyzer at a resolution of 35,000. TPs were identified with Compound Discoverer (version 3.3) and Xcalibur software (version 3.1.66.7) (Thermo Fisher Scientific) based on the molecular formula, mass accuracy (±5 mg/L), degree of unsaturation of ions, and MS/MS data.

#### 2.8.7. Toxicity Assessment

The toxicity of the samples taken at the end of 10 sequencing batches was measured using the Microtox test. This assay allows monitoring of the natural emissions (in the range of visible light, with a maximum intensity at 490 nm) of the marine bioluminescent bacterium *Vibrio fischeri* after exposure to selected samples. Toxicity data, corresponding to the 50% effective concentration (EC50), were based on a 5 and 15 min incubation of bacteria with filtered diluted samples (pH 7) at 25 °C. Toxicity was expressed as toxicity units (TU), which were calculated as TU = 100/EC50.

#### 2.8.8. Microbiological Analysis

##### DNA Extraction and PCR-DGGE Analyses

Microbial biomass from woodchips was extracted by using a sterile phosphate buffer (pH 7.0) consisting of KH_2_PO_4_ 5.3 g/L and K_2_HPO_4_ 10.6 g/L in an ultra-pure water solution. The woodchips were submerged in that solution inside sterile bags, agitated at 200 rpm for 30 min, and sonicated in a Selecta ultrasound bath for 7 min at 40 kHz. The biomass suspension was then centrifuged, and the pellets were preserved at −20 °C. Finally, the dry mass from the lignocellulosic support was determined by heating woodchips until weight stabilization occurred.

Total DNA was extracted by using the DNA DNeasy^®^ PowerSoil^®^ Pro Kit from Quiagen (Hilden, Germany), following instructions provided by the manufacturer. The total DNA of each sample was quantified using a Qubit 3.0 fluorometer and the dsDNA High-Sensitivity Assay Kit from Thermo Fisher Scientific per the instructions given by the manufacturer. DNA extractions for each condition were performed in triplicate.

A nested PCR was performed to amplify the ITS region of the DNA using fungal-specific primers to improve specificity and sensitivity. The primer sets EF4/ITS4 [39] and GC-ITS1F/ITS2R [40] were used for the first and second round, respectively. For the amplification of the V3-V4 region of 16S rDNA, a touch-down PCR was carried out with the bacterial-specific primers GC-341F and 907RM [41,42]. The primers GC-ITS1F and GC-341F contained a 40 bp GC-clamp at the 5′ end because the PCR products would later be resolved on a DGGE gel. The PCR mix contained 1X PCR Rxn Buffer, 1.5 mM MgCl_2_, 0.5 µM of each primer, and 0.05 U/µL *Taq* DNA Polymerase (Invitrogen, Thermo Fisher Scientific, USA). Cycling conditions for each PCR can be found in Appendix A. All PCR reactions were carried out in a Biometra T-Personal 48 thermal cycler. PCR products were checked in a 1.5% agarose gel in TBE buffer (Tris borate EDTA: TRIS base, 89 mM; boric acid, 89 mM; and EDTA, 2 mM) and quantified via image analysis using Quantity One 1-D analysis software version 4.6.9. The Low DNA Mass Ladder from Invitrogen™ was used as a standard for quantification on agarose gels.

Denaturing gradient gel electrophoresis (DGGE) was used as a fingerprint method for the analysis of shifts in the fungal and bacterial communities at the beginning and the end of the performance of three different reactors, including the initial stage of the colonized wood as previously described in Section 2.1. Fungal and bacterial amplicons were loaded on 6% (*w*/*v*) acrylamide gels (37.5:1 acrylamide-N:N′-methylene-bis-acrylamide) with 15-55% and 30–70% denaturing gradients, respectively. The 100% denaturing solution contained 7 M urea (Bio-Rad, Hercules, CA, USA) and 40% (vol/vol) deionized formamide (Merck). A total of 1200 ng of each sample was loaded on the gel and run through the TAE 1X buffer (40 mM Tris-acetate [pH 7.4], 20 mM sodium acetate, and 1 mM EDTA) for 16 h at 60 °C. Electrophoresis was carried out with the Dcode Universal Mutation Detection System (Bio-Rad), and gels were then stained with 0.5 ppm of ethidium bromide and photographed in a Bio-Rad Gel Doc system.

Prominent bands, excised from the gel and purified, were reamplified and sequenced by Macrogen, Inc. (Seoul, Republic of Korea). Sequences were manually trimmed and quality-checked using FinchTV 1.4.0 (Geospizza, Inc., Chicago, IL, USA). The 16S rRNA and ITS sequences were assigned to their closest neighbor according to the Basic Local Alignment Search Tool (BLAST) results [43]. Curated sequences were deposited in the National Center for Biotechnology Information (NCBI) GenBank database under accession numbers PP694412-PP694484 for bacteria and PP693959-PP694045 for fungi.

##### Quantitative Real-Time PCR (qPCR)

A quantitative PCR of total bacteria, total fungi, and specifically the fungal genus *Ganoderma* immobilized on oak wood chips were carried out with the primer sets Com1/769R [44], ITS3/ITS4 [45], and ITS1F/Gano2R [46], respectively. qPCR assays were performed in a Bio-Rad CFX96™ Real-Time System C1000™ Thermal Cycler controlled by Bio-Rad CFX Manager software version 3.1 using the cycling conditions described in Appendix A. Each reaction mixture, with a final volume of 20 µL, contained 1X ssoAdvanced Universal SYBR Green Supermix (Bio-Rad); 1 µM, 0.5 µM, and 0.5 μM of each forward and reverse primer for total bacteria, total fungi, and *Ganoderma* sp., respectively; and 36 ng of DNA for most samples. All reactions were run in triplicate. Calibration curves were prepared with known amounts of *Salmonella* sp., *Trichoderma hazenarium*, and *G. lucidum.* Negative target controls using *Trichoderma hazenarium, Pycnoporus sanguineus*, and *Saccharomyces cerevisiae* were used in the case of *Ganoderma* sp. quantification to ensure there was no unspecific amplification.

#### 2.8.9. Data Analysis

The removal percentage was calculated using Equation (1).(1)$$FR\;removal\;percentage=\frac{\left(C_{0}-C_{t}\right)}{C_{0}}\cdot 100\%$$
where C_0_ and C_t_ are the initial and residual FR concentrations (mg/L) in the culture at t_0_ and at a given time t (d), respectively. The mean and standard deviation (SD) of triplicate measurements were calculated.

To assess the performance of FR biodegradation, it is necessary to perform a mass balance of the pollutants (Equation (2)).(2)$$FR\;biodegradation\;percentage=\frac{\left(C_{0}V-(C_{L}V+C_{F}X)\right)}{C_{0}V}\cdot 100\%$$
where C_0_ is the initial FR concentration at t_0_ (mg/L), C_L_ is the residual FR concentration in the liquid phase (mg/L), and C_F_ is the FR concentration in the biomass (mg/g DW) at the end. V is the treated volume (L) and X is the biomass dry weight (g).

DGGE profiles described in Section 2.8.8 were normalized and analyzed with InfoQuest™ FP version 4.5. The number and relative intensity of all DGGE bands for each genetic profile were determined, and hierarchical cluster analysis was performed using the UPGMA (unweighted pair-group method with arithmetic means) method on a Dice distance matrix. Furthermore, the relative abundance of each taxonomical group, as well as Shannon’s diversity index (H) [47] and Pielou’s evenness index (E) [48], were calculated from the intensity matrix. Data mean and standard deviation (SD) were calculated and subjected to statistical significance determination using SPASS v22.0 and GraphPad Prism 8.

#### 2.8.10. Other Analyses

Glucose concentration was measured using a biochemistry analyzer (2700 select, Yellow Spring Instrument, Yellow Springs, OH, USA) after filtering the sample with a Millipore Millex-GV PVDF syringe filter (0.22 µm) (Merck Millipore, Burlington, VT, USA).

Laccase activity was measured through the oxidation of DMP by the enzyme as described elsewhere [49]. After, the sample was filtrated by using a 0.22 μm hydrophilic polypropylene syringe filter (Scharlau, Barcelona, Spain). Activity units per liter (U/L) are defined as the amount of DMP in μM which is oxidized in one minute. The molar extinction coefficient of DMP was 24.8/(mM/cm). Glucose concentration was measured using a biochemistry analyzer (2700 select, Yellow Spring Instrument, USA) after filtering the sample with a Millipore Millex-GV PVDF syringe filter (0.22 µm).

The absorbance at a wavelength of 650 nm was determined by a UNIGAM 8625 UV/VIS spectrometer (Manchester, UK) to detect color.

COD was analyzed using commercial kits such as LCK 314, LCK 114, and LCK 514 (Hach Lange, Düsseldorf, Germany).

## 3. Results

### 3.1. Screening of Fungal Strains for OPFR Removal in Erlenmeyer Flasks

#### 3.1.1. Degradation of TCEP and TBP by Different WRF

The profiles of the TBP relative concentrations obtained from the experimental treatment, killed control, and abiotic control with *T. versicolor*, *G. lucidum*, and *P. velutina* are shown in Table 1. Note that TBP exhibited high chemical stability in the abiotic controls during a 1-week incubation, indicating that any removal observed in the other two biotic experiments can be attributed exclusively to adsorption and biodegradation. Up to 37% and 8% of TBP was removed in culture-killed control experiments using the autoclaved biomass of *G. Lucidum* and *P. velutina*, respectively, showing a significant adsorption contribution. Nevertheless, 100% and 90% removals were obtained in the experimental flasks with *G. lucidum* and *P. velutina*, respectively, demonstrating the important role played by the degradative activity of both strains. A maximum laccase activity (6.31 U/L) was observed on the seventh incubation day of *G. lucidum,* while this activity was around 0.1 on the third day of the experiment. In the case of *T. versicolor*, laccase activity reached the maximum (8.46 U/L) on the third day and then decreased to 3.74 on the seventh day. Glucose concentration plunged to almost zero after 3 d.

Ergosterol, the main sterol in fungal cell membranes, was employed as fungal biomass indicator [50]. Ergosterol concentration increased over time in all strains up to 30 days, where it was 0.136 ± 0.049, 0.103 ± 0.006 and 0.083 ± 0.004 mg/gdry wood for *G. lucidum*, *T. versicolor* and *P. velutina*, respectively.

Based on the results shown in Table 1, *T. versicolor* and *G. lucidum* were selected to further explore the degradation capacity of OPFRs. Another experiment evaluated the degradation of TCEP by *G. lucidum* and *T. versicolor* (as shown in Appendix A). The latter was also studied, given its proven ability to degrade similar pollutants [51]. In this case, the removals achieved were less than 30% for both strains, indicating that this compound is less susceptible than TBP to fungal bioremediation. Laccase activity was relatively low for both strains, although this is not indicative of fungal inactivity. Therefore, no firm connection between biodegradation and laccase can be proposed based on these results. Similar behavior has been observed when dealing with other pollutants [33,52].

Since the results of the batch degradation experiments revealed that biomass adsorption contributed significantly to the overall removal process, global mass balances covering the entire culture system were performed to determine the final fate of the adsorbate. As presented in Table 2, trace amounts of TBP of less than 0.2% were detected as residues in the solid and liquid phases. Thus, removal can be mainly attributed to fungal biodegradation. In the case of TCEP, the mass balance results confirmed the low removal achieved by both adsorption and biodegradation mechanisms.

#### 3.1.2. Contribution of Cytochrome P450 to the Degradation of TBP and TCEP by *G. lucidum*

The activity of the CYP450 enzymatic system in TBP and TCEP removal was assessed using in vivo degradation experiments following the addition of the cytochrome P450 inhibitor 1-aminobezotriazole. As shown in Figure 1A, the degradation rate of TBP decreased significantly in the presence of the CYP450 inhibitor (i.e., ABT), compared to the inhibitor-free condition, with final elimination percentages being 40.1 and 83.4% after 45 h, respectively. Removal of TCEP, on the other hand, showed a percentage of 1.2 and 28.2% after 15 days (Figure 1B).

#### 3.1.3. Transformation Products Generated During TBP and TCEP Degradation

Regarding TPs, a total of six compounds, four from TBP and two from TCEP, were identified over 7 days of treatment by UHPLC-MS/MS analysis. The TPs can be found in Appendix A for TBP and TCEP, respectively. Dibutyl 3-hydroxy butyl phosphate (OH-TBP), dibutyl phosphate (DBP), butyl dihydrogen phosphate (MBP), and butyl 3-hydroxy butyl phosphate (OH-DBP) were identified from TBP degraded by *G. lucidum* and *T. versicolor*, while bis(2-chloroethyl) 2-hydroxyethyl phosphate (TCEP-OH) and 6-chloro-5-oxohexyl dihydrogen phosphate were detected in the TCEP degradation experiment for both the *T. versicolor* and *G. lucidum* strains.

### 3.2. TCEP and TBP Removal in a TBR Under Non-Sterile Conditions

#### 3.2.1. Degradation of TCEP and TBP

Based on the degradation studies in Erlenmeyer flasks (Section 3.1), *G. lucidum* was chosen for the degradation of the TCEP and TBP mixture in a TBR operating in batch sequences under non-sterile conditions. Figure 2 shows the relative removals for both compounds. In the first batch, TBP and TCEP removals were considerably high, being above 90% and 80%, respectively. However, in the subsequent batches, the TBR removal capacity decreased progressively, especially in the case of TCEP. In this regard, average eliminations of 85.3% of TBP and 54.8% of TCEP were achieved after 10 cycles.

The marginal difference between the killed control and experimental reactors (Figure 2) indicates that the main removal mechanism for both compounds in the killed control reactor is adsorption. Furthermore, although both the experimental and killed control reactors operated on a 2 day cycle, the actual contact time between the fungi and pollutants in each cycle was relatively short. The contact time represents the average duration for the liquid to pass through the reactor, which was measured to be 15.96 ± 1.65 s. With a recirculation flow rate of 200 mL/min, the treated wastewater (1 L) circulates 576 times through the column per cycle, which means a total contact time of 153.2 ± 15.8 min.

The TBP and TCEP monitoring results in the liquid phase at the end of each cycle suggested a significant contribution of adsorption, which was further verified through the analysis of the remaining pollutants in the solid phase and the application of the mass balance after 10 cycles (Table 3 and Table 4). Despite the remarkable contribution of adsorption in all the reactors (especially for TCEP), the killed control reactors showed better results than the control reactors in terms of the adsorbate amount remaining in the solid phase, evidencing a positive effect of *G. lucidum* biomass on pollutant adsorption. In the killed control reactor, the amount of contaminant detected was higher compared to that of the abiotic control, indicating that the film of the heat-inactivated fungus increased the adsorption effect. The highest global degradation percentage for both contaminants was detected in the abiotic reactor throughout the 20-day operation period. Furthermore, the 40.91% of concentration of untreated wastewater causes a reduction of 50% in the natural emissions of the bioluminescent bacterium *Vibrio fischeri*, while after treatment in the last cycle in all reactors the effect on the emission was less than 50%. Consequently, toxicity was reduced from 2.4 TU in the untreated wastewater to 0 after treatment in the last cycle in all reactors, including in the abiotic reactor, which was mainly attributed to adsorption.

Chemical oxygen demand (COD) and turbidity were monitored at the end of each batch. A stepwise reduction in COD was observed from the first to the ninth cycle in all setups, with higher COD residues in the effluent of the abiotic control compared to the experimental and killed control treatments from the third cycle onward. However, a sudden increase in COD was detected in the experimental and killed control reactors during the 10th cycle (Appendix A), which could be attributed either to the addition of nutrients [24] or the release of organic matter from biomass [53] due to fungal cell autolysis or wood decomposition by *G. lucidum*. The latter may also explain the differences between the experimental and killed control reactors. A consistent decrease in turbidity was observed throughout the experiment.

#### 3.2.2. Characterization of Microbial Populations Immobilized on the Lignocellulosic Support

The fungal DGGE profiles (Appendix A) revealed between 5 and 12 detectable bands per lane. A total of 100 bands were recovered from the gel, and after purification, reamplification, sequencing, trimming, and checking, 87 were selected to upload. Overall, these sequences fell into two different phyla (*Basidiomycota* and *Ascomycota*), seven classes (*Agaricomycetes*, *Cystobasidiomycetes*, *Eurotiomycetes*, *Microbotryomycetes*, *Saccharomycetes*, *Sordariomycetes*, and *Tremellomycetes*), and eight orders (*Cystobasidiales*, *Euroriales*, *Filobasidiales*, *Polyporales*, *Saccharomycetales*, *Sordariales*, *Sporidiobolales*, and *Trichosporonales*) (Appendix A). With reference to bacterial DGGE fingerprints (Appendix A), the gel exhibited between 4 and 11 detectable bands per lane. The total number of bands recovered was 103, but after purification, reamplification, sequencing, trimming, and checking, 73 were selected to upload. These sequences fell into four different phyla (*Proteobacteria*, *Actinobacteria, Acidobacteria*, and *Bacteroidetes*), six classes (*Alphaproteobacteria*, *Betaproteobacteria*, *Gammaproteobacteria*, *Actinomycetia*, *Sphingobacteriia*, and *Terriglobia*), and nine orders (*Burkholderiales*, *Hyphomicrobiales*, *Lysobacterales*, *Micrococcales*, *Mycobacterales*, *Rhodospirillales*, *Sphingobacteriales*, *Sphingomonadales*, and *Terriglobales*) (Appendix A). Band positions and phylogenetic affiliations, obtained after comparison with closely related GenBank sequences, can be seen in Appendix A and Appendix A for fungi and bacteria, respectively.

The dendrograms generated from the hierarchical clustering analysis of the DGGE banding patterns (Appendix A) revealed distinct differences in both fungal and bacterial compositions among the experimental treatment (EX) and the control reactors (AC and KC), as well as in comparison to the initially colonized wood. The fungal community dendrogram (Appendix A) showed two primary clusters with an 81.74% dissimilarity. The first cluster grouped the experimental treatment with the initially colonized wood (61.63% dissimilarity), reflecting the dominance of *G. lucidum* in the fungal biomass immobilized on the woodchips, as this species was originally used to colonize the lignocellulosic substrate. The second cluster grouped both control reactors (63.51% dissimilarity) with the fungal diversity primarily influenced by the wastewater influent. The differences in colonization patterns between both controls can be attributed to the fact that, in the killed control, the woodchips were colonized by the heat-inactivated *G. lucidum* mycelium, while in the abiotic control, the wood lacked any immobilized organic matter. The dendrogram for bacterial communities (Appendix A) showed consistent trends. In this case, the experimental treatment was grouped with both controls, while the initially colonized wood was distinctly separated with an 82.59% dissimilarity. This was expected, as the bacterial populations that colonized the woodchips in all three reactors (EX, AC, and KC) originated from the influent. Within this main group (60.32% dissimilarity), both control reactors were more closely related to each other (44.75% dissimilarity) and differed from the experimental treatment. This difference is likely due to the presence of pre-established *G. lucidum* in the experimental reactor, which could have influenced the subsequent colonization of the lignocellulosic substrate by native synthetic wastewater bacteria.

As expected, the fungal community analysis in Figure 3A revealed a total predominance of *G. lucidum* on the initially colonized wood and a high prevalence on the experimental treatment (77.12 ± 0.01%). *Rhodotorula* sp. partially colonized both control bioreactors but with different success depending on the conditions, representing 89.47 ± 0.02% and 30.40 ± 0.01% in the AC and KC, respectively. The high abundance of *Rhodotorula* sp. caused the abiotic control diversity to be the lowest. Otherwise, its presence was residual in the experimental bioreactor since *Rhodotorula* sp. had to compete with the settled *G. lucidum*. In the abiotic control, the fungal populations from the synthetic wastewater could colonize woodchips without competing with any pre-existing attached fungal biomass. However, there was a clear pressure exerted by the added OPFRs. This pressure would also exist on the killed control, but in this case, the inactivated *G. lucidum* could serve as an extra carbon source, which seemed to prepare the substrate to host a wider diversity of fungi.

Regarding the bacterial community (Figure 3B), an increase in diversity was observed during synthetic wastewater treatment as evidenced by comparing the Shannon diversity index of the initially colonized wood (CW) with those corresponding to the experimental treatment (EX) and the killed (KC) and abiotic (AC) controls at the end of the experiment. In the initially colonized wood (CW), the presence of *Rhodococus* sp., *Burkholderia* sp., and *Achromobacter* sp. was observed, representing 43%, 20.8%, and 26.1% of relative abundance. However, at the end of the treatment, *Rhodococcus* sp. was not detected in any of the bioreactors, while *Burkholderia* sp. and *Achromobacter* sp. were only detected at low concentrations in the killed control, where they showed relative abundances of 4% and 6.3%, respectively. Therefore, these initial bacterial populations were replaced by more complex bacterial assemblages, where *Terriglobus* sp. (40.6%), *Rhizobium* sp. (31.4%), and *Novosphingobium* sp. (24.1%) along with *Terriglobus* sp. (24.3%) were the most abundant in the experimental reactor (EX) and in the killed and abiotic controls, respectively.

Quantification by qPCR (Figure 4) corroborated the DGGE results, revealing that *G. lucidum* remained stable on the support after 10 cycles in the bioreactor, dropping only by 1 order of magnitude. The bacterial populations, as expected, increased after the colonized wood was exposed to the synthetic wastewater. Regarding fungal populations, fungi in the influent water managed to colonize wood chips from both controls quite efficiently since their concentration exceeds that of the initially colonized wood. Furthermore, *G. lucidum* was not detected in either control. Calibration curves, efficiency, and R^2^ values can be checked in Appendix A.

## 4. Discussion

TBP has been reported to be successfully removed by different bacterial [54,55] and fungal strains [50,56]. However, this study confirms that *T. versicolor*, *G. lucidum*, and *P. velutina* are excellent TBP degraders, thus enriching the list of candidate organisms that can be used for bioremediation. A recent screening of WRF as potential degraders of several OPFRs suggest that polarity correlates with its susceptibility to fungal degradation [57]. The absence of laccase activity produced by *P. velutina* suggests a limited involvement of this enzyme in TBP degradation, as previously hypothesized for certain OPFRs elsewhere [56]. On the contrary, the presence of ABT in the culture notably reduced TBP and TCEP degradation compared to an inhibitor-free culture system. However, a small amount of removal was observed initially, likely due to the sorption process. These results align with a previous study demonstrating that ABT significantly suppresses TBP degradation and its metabolite formation in *T. versicolor* [58], and consequently, the intracellular enzymatic cytochrome P450 is involved in the degradation of OPFRs [57].

Considering the identified transformation products by both fungi, and according to their formulas, it can be hypothesized that OH-TBP was produced by the first hydroxylation of TBP, which is generally the first detoxification step in eukaryotic cells and mammals to enhance hydrophilicity and bio-accessibility of pollutants [59]. This is in line with the finding that cytochrome P450 is significantly involved in TBP degradation. MBP can be metabolized through two different pathways: by double hydrolysis of OH-TBP catalyzed by phosphodiesterases or directly from TBP through stepwise excision of the ester bonds [60]. The identified transformation products for TBP allowed us to recently propose a degradation pathway by *T. versicolor* [58]. The TCEP-derived TPs are also consistent with those previously reported for microbial degradation, which are suspected of following phosphodiester bond hydrolysis, hydrochlorination dechlorination, and oxidation mechanisms [61]. Considering that enzymatic system tests identified the CYP450 intracellular system responsible for OPFR degradation, reactions of hydroxylation, dealkylation, and dehalogenation are possibly involved in the degradation pathway [57]. Nevertheless, further investigation is still needed to elucidate the entire metabolic pathway for TCEP.

Although G. lucidum has demonstrated a high and limited capacity to degrade TBP and TCEP, respectively, in sterile conditions, the treatment of synthetic wastewater spiked with both pollutants in a trickle-bed reactor with the fungus immobilized on wood chips and under non-sterile conditions showed an important contribution of the adsorption process in pollutant removals. This fact is ascribed to the wood chips, which are generally porous materials that can contribute significantly to the overall removal in fungal treatments [37,62]. The higher affinity of TBP for wood was mainly attributed to the more hydrophobic nature of this compound (log K_ow,TBP_ = 4.00; log K _ow,TCEP_ = 1.78) [63]. In addition, the short contact time between the fungal biomass and the wastewater flowing through the 0.5 m trickle-bed reactor (2.55 h) may have constrained the enzymatic activity of *G. lucidum*, particularly cytochrome P450, in degrading TCEP [64]. Consequently, the restricted contact time in the trickle-bed reactor likely hindered the full potential of the fungal degradation process in the experimental setup, impacting the overall efficiency of OPFR removal.

The characterization of the fungal populations immobilized on the lignocellulosic support evidenced that *Rhodotorula* sp. could play an important role in the degradation of TBP and TCEP in both controls since members of this genus have been reported to exhibit great degrading capabilities for compounds that are similar to the pollutants studied [65,66]. The killed control also presented a significant abundance of *Apiotrichum* sp., *Naganishia* sp., and *Chaetomium* sp. The latter also could participate in the degradation of the spiked flame retardants [67,68].

Several species detected during the analysis of the bacterial community, such as *Rhizobium* sp., *Luteibacter* sp., *Novosphingobium* sp., and *Sphingobacterium* sp., have degrading capabilities of similar compounds [69,70,71,72] and, therefore, could contribute to the degradation of the added pollutants in both controls where they were more prominent, and likely played a less significant role in the experimental treatment where *Ganoderma* was the most relevant OPFR degrader. It is remarkable that *Novosphingobium* sp. was only encountered on the abiotic control where *Rhizobium* sp. was not detected; this could also be explained by the fact that both genera could occupy a similar niche on the bioreactor, competing for the same resources, which would likely be the added OPFRs. However, as mentioned earlier, these differences in the composition of bacterial populations could also be due to the fact that the inactivated *G. lucidum* acts as an extra carbon source. In addition, *Terriglobus* sp. was abundant in all three bioreactors but was absent in the initially colonized wood, so it is likely to have a high abundance on the influent wastewater or strong compatibility with the support material. In contrast, *Rhodococcus* sp., which has OPFR-degrading capabilities [73], exhibited a high relative abundance on the initial colonized wood but was completely lost in the three reactors and was replaced by other microbial populations with better fitness to survive under those stressful conditions and that could use the available resources. On the other hand, diversity indices obtained from the bacterial DGGE profiles did not show significant differences among the three reactors. Still, as expected, these were higher than those corresponding to the initially colonized wood.

In summary, the characterization of the microbial assemblage immobilized on the lignocellulosic support showed that *Ganoderma lucidum* is the most abundant fungus and the main one responsible for the degradation of TCEP and TBP in the experimental bioreactor. However, these analyses also revealed the presence of another fungus, *Rhodotorula* sp., and several bacterial populations such as *Rhizobium* sp., *Luteibacter* sp., *Novosphingobium* sp., and *Sphingobacterium* sp. All of them have great potential to degrade the OPFRs, and while their abundance was low or non-existent in the experimental bioreactor, it increased significantly in the killed and abiotic control bioreactors. Consequently, these wastewater-indigenous bacterial and fungal populations have a relevant role in removing TBP and TCEP in both control bioreactors.

## 5. Conclusions

This study has demonstrated the potential of *G. lucidum* for the effective biodegradation of OPFRs, specifically TCEP and TBP, under non-sterile conditions in a TBR. The results revealed that *G. lucidum* immobilized on oak wood chips achieved significant removal efficiencies, with an average elimination of 85.3% for TBP and 54.8% for TCEP over 10 operational cycles. The enhanced removal of TBP, compared to TCEP, can be attributed to the more hydrophobic nature of TBP, which facilitated greater adsorption onto the porous wood substrate, as well as the relatively low contact time in the TBR. Although adsorption contributed substantially to the removal of both compounds, the experimental reactors exhibited a high performance compared to the control reactors, confirming the active role of *G. lucidum* in pollutant degradation. The intracellular enzymatic system cytochrome P450 is involved in the degradation of both TBP and TCEP. Microbial community analysis revealed that *G. lucidum* maintained its dominance throughout the TBR operation despite the non-sterile conditions, indicating its resilience and suitability for real-world wastewater treatment environments. Furthermore, *G. lucidum*’s cooperation with indigenous bacterial and fungal populations in synthetic wastewater contributed to enhanced degradation, as microbial interactions likely supported the biodegradation process. These findings highlight the capability of fungal-based bioreactors as a scalable, sustainable, and efficient solution for the bioremediation of recalcitrant pollutants, offering a viable alternative to conventional wastewater treatment methods.

## Figures and Tables

**Figure 1 jof-11-00085-f001:**
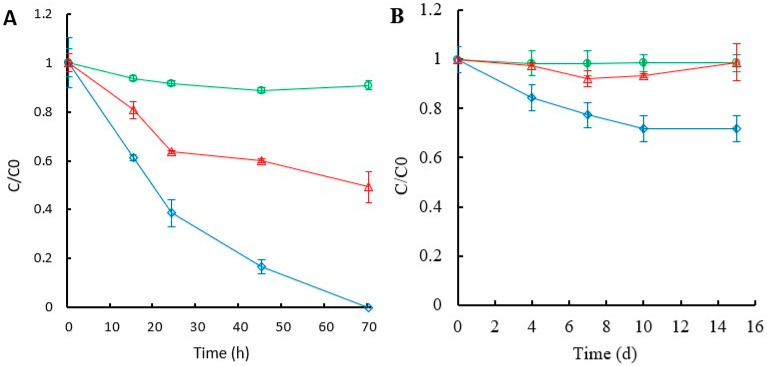
Contribution of the CYP450 system in TBP (**A**) and TCEP (**B**) degradation. Degradation profiles of *G. lucidum* in the presence and absence of the ABT inhibitor. Empty rhombus (colored in blue), inhibitor free; empty triangle (color in red), in the presence of the inhibitor; empty circle (color in green), abiotic control. C and C_0_ represent residual and initial concentration, respectively. Average values of three replicates with the corresponding standard deviations are shown.

**Figure 2 jof-11-00085-f002:**
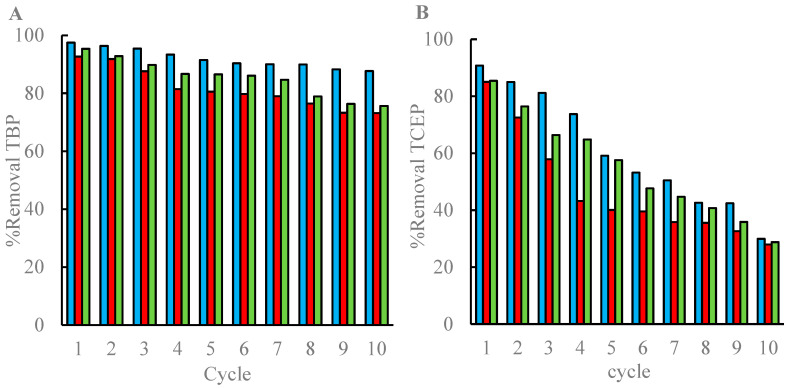
Profiles of TBP (**A**) and TCEP (**B**) removals by *G. lucidum* in a TBR under non-sterile conditions. The removals are presented as follows: blue columns are the abiotic control (AC), red columns are the killed control (KC), and green columns are the experimental (EX) treatment.

**Figure 3 jof-11-00085-f003:**
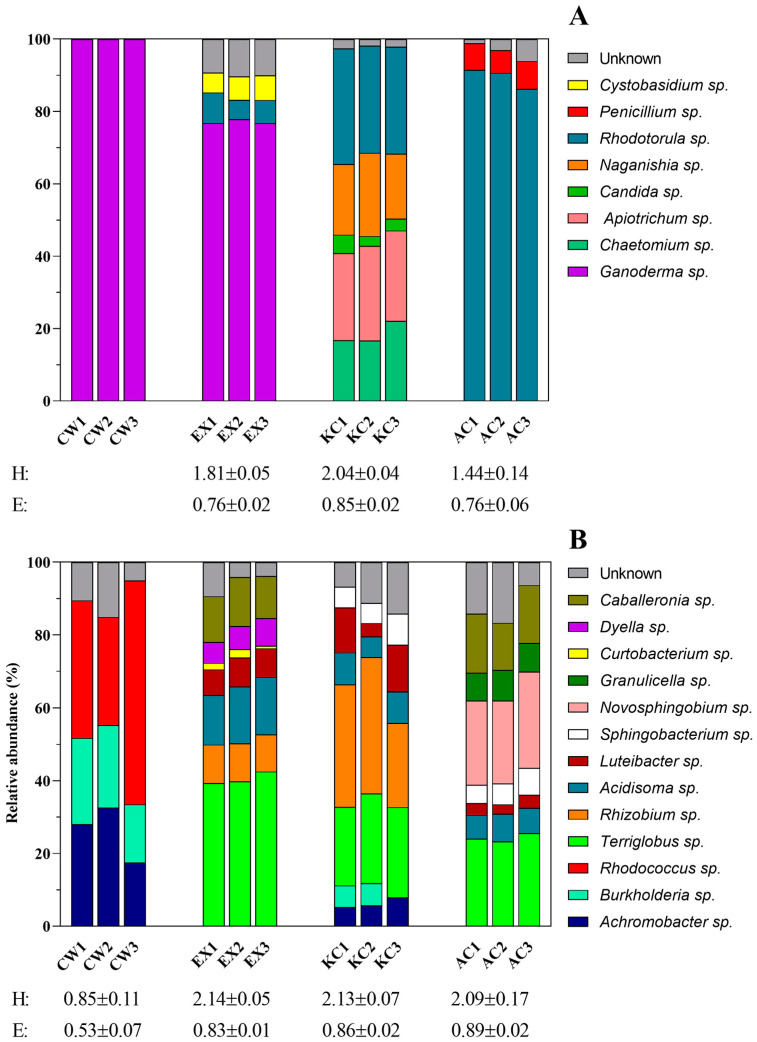
Relative abundance of the fungal (**A**) and bacterial (**B**) taxonomical groups at the genus level in the immobilized biomass from the wood chips packed in the trickle-bed reactor corresponding to the abiotic control (AC), killed control (KC), and experimental treatment (EX) in comparison with the initial colonized wood (CW) (left column); three replicates are represented for each condition. Below each sample, Shannon’s diversity index (*H*) and Pielou’s evenness index (E), which were calculated from the DGGE intensity matrix, are indicated.

**Figure 4 jof-11-00085-f004:**
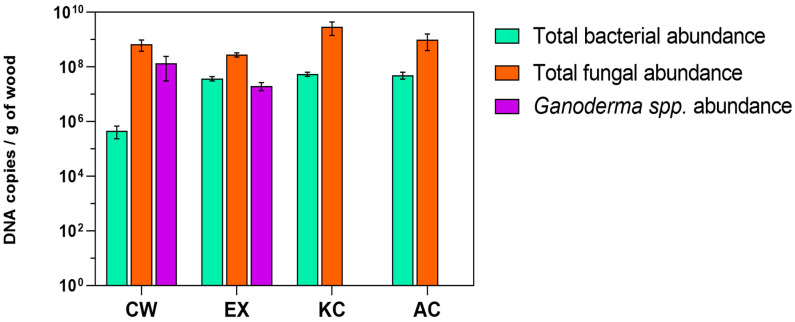
The abundance of *Ganoderma* spp. and total fungal and bacterial populations on the wood chips packed in the trickle-bed reactor corresponding to the abiotic control (AC), killed control (KC), and experimental treatment (EX) after a 20-day operation, in comparison with the initially colonized wood (CW), expressed as the log of DNA copies of the qPCR target amplicon by the dry mass of wood in grams.

**Table 1 jof-11-00085-t001:** Time course of percentage of residual TBP concentration during *T. versicolor*, *G. lucidum*, and *P. velutina* degradation. C and C_0_ represent the residual and initial TBP concentration (mg/L), respectively.

C/C_0_ (%)	*T. versicolor*		*G. lucidum*		*P. velutina*	
	t: 3 d	t: 7 d	t: 3 d	t: 7 d	t: 3 d	t: 7 d
Experimental treatment (EX)	1.9 ± 0.4	0.0	4.2 ± 0.4	0.0	17.1 ± 2.9	10.6 ± 1.2
Killed control (KC)	70.3 ± 10.7	65.8 ± 11.3	68.9 ± 4.8	63.4 ± 2.5	92.9 ± 0.2	92.3 ± 1.0
Abiotic control (AC)	98.7 ± 1.4	97.5 ± 1.5	99.9 ± 2.3	99.7 ± 2.9	99.7 ± 0.2	98.9 ± 0.5

Note: Mean and standard deviations are shown.

**Table 2 jof-11-00085-t002:** Mass balance profile of TBP and TCEP in Erlenmeyer flasks after 7 d of incubation.

Item	Mass (ng)			
	*G. lucidum*		*T. versicolor*	
	TBP	TCEP	TBP	TCEP
Introduced	657,635	978,513	657,635	978,513
Remained in solid phase	38 ± 3	47,309 ± 11,027	312 ± 25	17,732 ± 2954
Remained in liquid phase	873 ± 11	716,667 ± 27,813	521 ± 255	688,168 ± 51,371
Degradation	656,724 ± 14	214,537 ± 38,840	656,802 ± 280	272,613 ± 54,325
Removal (%)	99.9%	26.8%	99.9%	29.7%
Degradation (%)	99.9%	21.9%	99.9%	27.9%

Note: Mean and standard deviations are shown.

**Table 3 jof-11-00085-t003:** Mass balance of TCEP after 10 cycles of batch treatment by *G. lucidum* in a TBR under non-sterile conditions.

TBR	Mass of TCEP (mg)			Adsorption (%)	Degradation (%)
	Initial Mass in the Liquid Phase	Final Mass in the Solid Phase	Final Mass in the Liquid Phase		
Experimental	100	68.5	22	68.5	9.5
Killed control	100	75.3	24.7	75.3	0
Abiotic control	100	50.0	18.7	50.0	31.3

**Table 4 jof-11-00085-t004:** Mass balance of TBP after 10 cycles of batch treatment by *G. lucidum* in a TBR under non-sterile conditions.

TBR	Mass of TBP (mg)			Adsorption (%)	Degradation (%)
	Initial Mass in the Liquid Phase	Final Mass in the Solid Phase	Final Mass in the Liquid Phase		
Experimental	100	19.5	23.8	19.5	56.7
Killed control	100	37.5	28.3	37.5	34.2
Abiotic control	100	24.7	12.4	24.7	62.9

## Data Availability

The original contributions presented in this study are included in the article/Appendix A. Further inquiries can be directed to the corresponding author.

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
