# Peer review of "Ganoderma lucidum* Immobilized on Wood Demonstrates High Persistence During the Removal of OPFRs in a Trickle-Bed Bioreactor"

_jof, 2025, doi:10.3390/jof11020085_

Round 1

Reviewer 1 Report

The manuscript under review presents a comprehensive study of organophosphorus flame retardants by white rot fungi. The potential of WRFs as a potential biodegradants of OPFRs is revealed. OPFRs pose a serious threat to human health, so the significance of the study is beyond doubt. However, the manuscript requires some corrections before publication.

Major comments:

1.       1. The description and presentation of TCEP and TBP degradation in TBR needs to be improved. The current presentation of the data makes it difficult to elucidate the role of G. lucidum in OPFR biodegradation. As can be seen from Figure 2, the highest removal rate was achieved in the abiotic controls. I believe this was due to colonization of the wood chips by other microorganisms along with adsorption? This part needs to be discussed thoroughly. I also propose to add the values of biodegraded OPFRs to Table 3 to highlight the role of biodegradation in these experiments.

2.       The MM section lacks some methods: measurements of laccase activity and desorption procedures both in flask experiments and in TBR experiments.

3.       In my experience, MeOH can affect laccase production in WRFs. What was the MeOH concentration in the flask experiments? Was a control setup with MeOH and no OPFR used?

4.       The manuscript lacks sufficient discussion of the results obtained in this work in the context of another studies on the biodegradation of these compounds.

Minor comments:

Table 1. Please, consider using % of residual TBP.

Table 2. I suppose it would be better to round the values (and remove decimals).

Figure 1. Is it possible to add the heat killed control curve to the graph? I think it might improve the presentation of the data.

L607-608: Is this true? If the Figure 2 shows the % of OPFRs removed from the liquid, then the abiotic control reactor seems to be the best.

Please, check the italics in fungal names throughout the manuscript.

Perhaps the abbreviations that are used only in the introduction can be omitted.

L16-17: shortened names can be omitted there.

I also suggest adding a photo or scheme of the reactor in the supplementary.

Author Response

We appreciate the effort in reviewing our manuscript.

Find enclosed responses to your questions and comments in the atacked file.

The manuscript under review presents a comprehensive study of organophosphorus flame retardants by white rot fungi. The potential of WRFs as a potential biodegradants of OPFRs is revealed. OPFRs pose a serious threat to human health, so the significance of the study is beyond doubt. However, the manuscript requires some corrections before publication.

Thank you very much for taking the time to review this manuscript. Please find the detailed responses below. The manuscript has been changed significatively because we did not separate the results section and the discussion section. As consequently the changes marked were too much to the readable. So, we include the revised version in the clean form but if reviewer prefer with changes marked version, we can provide it.

Thank you for your positive comments and your suggestions to improve our manuscript.

Major comments:

  1. The description and presentation of TCEP and TBP degradation in TBR needs to be improved. The current presentation of the data makes it difficult to elucidate the role of G. lucidum in OPFR biodegradation. As can be seen from Figure 2, the highest removal rate was achieved in the abiotic controls. I believe this was due to colonization of the wood chips by other microorganisms along with adsorption? This part needs to be discussed thoroughly. I also propose to add the values of biodegraded OPFRs to Table 3 to highlight the role of biodegradation in these experiments.

Thank you for pointing this out.

The degradation capacity of Ganoderma lucidum is evidenced in the degradation experiments performed in Erlenmeyer flasks and sterile conditions. Nevertheless, in the application in the trickle bed bioreactor under non-sterile conditions and with a short contact time between the fungal biomass and the wastewater, the marginal difference between the killed control and experimental reactors indicate that main removal mechanism for both pollutants may be adsorption but mass balance along 10 cycles indicated certain degradation in all reactors because Ganoderma lucidum was the most predominant fungus present in the experimental treatment. Furthermore, the characterization of the microbial populations immobilized on the lignocellulosic support allowed the detection of another fungus, Rhodotorula sp., and several bacterial populations such as Rhizobium sp. Luteibacter sp. Novosphingobium sp. and Sphingobacterium sp. all of them with a great potential for degrading OPFRs. However, these populations were less abundant or absent in the experimental bioreactor. Therefore, taking into account the results obtained, and considering that previously, in section 3.1, the capacity of G. lucidum to degrade these contaminants was already demonstrated, we can conclude that G. lucidum is the main degrader in the experimental reactor. However, in killed and abiotic control bioreactors, the TBP and TCEP removal was carried out by the indigenous wastewater bacterial and fungal populations discussed above, which were more abundant in both control reactors.

Finally, it is important to highlight that adsorbed pollutant on wood can also be degraded in solid state fermentation as was demonstrated in recent publication, although extended time (3 months) is required.

Losantos D, Sarrà M(*), Caminal G (2024). Removal of TBP sorbed on wood by Trametes versicolor through solid-state fermentation. J. Hazard Mater. 480, 136066

https://doi.org/10.1016/j.jhazmat.2024.136066

Accordingly, to the comment, we have revised the discussion in order to clarify those aspects.

  1. The MM section lacks some methods: measurements of laccase activity and desorption procedures both in flask experiments and in TBR experiments.

We agree with the comment and consequently we have added to the materials and methods of manuscript the measurement of laccase (section 2.8.10) and the procedures for the OPFRs extractions (sections 2.8.4 and 2.8.5).

  1. In my experience, MeOH can affect laccase production in WRFs. What was the MeOH concentration in the flask experiments? Was a control setup with MeOH and no OPFR used?

We agree with the comment because MeOH can be even toxic, but our stock solution of the pollutant was 1000ppm, so for the degradation experiment the MeOH was 1% v/v. In this occasion we did not do the control, because is our common procedure with apolar pollutants. For example, during diuron degradation laccase was detected, although laccase was not able to transform diuron.

  1. The manuscript lacks sufficient discussion of the results obtained in this work in the context of another studies on the biodegradation of these compounds.

We apologise for the formal mistake of non-presenting the results and discussion in separated sections. In the revised version, we hope that the discussion is clearer.

Minor comments:

Table 1. Please, consider using % of residual TBP. Thank you for the suggestion. Now the values are clearer.

Table 2. I suppose it would be better to round the values (and remove decimals). We appreciate the suggestion. It is done.

Figure 1. Is it possible to add the heat killed control curve to the graph? I think it might improve the presentation of the data.

For the experiment set of Figure 2, killed control has not been added because it was done in the Table 1 set of experiments.

L607-608: Is this true? If the Figure 2 shows the % of OPFRs removed from the liquid, then the abiotic control reactor seems to be the best.

It is true but further explanation is included in the discussion section as was explained in the answer of the comment 1.

Please, check the italics in fungal names throughout the manuscript.

The names of fungi have been reviewed and corrected throughout the manuscript following the reviewer's suggestions.

Perhaps the abbreviations that are used only in the introduction can be omitted.

We agree with the comment and abbreviations only used one time have been removed.

L16-17: shortened names can be omitted there.  Done

I also suggest adding a photo or scheme of the reactor in the supplementary.

Done

Reviewer 2 Report

This article is an important contribution to the scientific discussion on this topic. The research is appropriately designed and the methods are described in detail. The results in the main article are well described and easy to interpret. Unfortunately, too many results that are relevant to the reader (e.g. transformation products, toxicity, COD, etc.) have been included in the supplement, making them difficult to analyse. More importantly, the authors did not include supplementary material during the submission of the manuscript, so it is difficult to evaluate the results obtained. I suggest some of these results be included in the main article. Unfortunately, in this form, I cannot accept the article for publication in JoF.

-no italics for Latin names (L121; L547)

- lack of suplementary material

Author Response

We appreciate the effort in reviewing our manuscript.

I am sorry you couldn't see the suplementary information.

This article is an important contribution to the scientific discussion on this topic. The research is appropriately designed and the methods are described in detail. The results in the main article are well described and easy to interpret. Unfortunately, too many results that are relevant to the reader (e.g. transformation products, toxicity, COD, etc.) have been included in the supplement, making them difficult to analyse. More importantly, the authors did not include supplementary material during the submission of the manuscript, so it is difficult to evaluate the results obtained. I suggest some of these results be included in the main article. Unfortunately, in this form, I cannot accept the article for publication in JoF.

Answer:

When the manuscript was submitted we included the supplementary material, as we considered it to be an important part of the manuscript, but in the main text we only included the more relevant figures and tables. We are very sorry that there were technical problems with the accessibility of this document for the reviewer.

Reviewer 3 Report

Dear authors,

Thank you for an interesting article, but in my opinion it needs significant revision. As I read it, I have a number of questions and comments.

1.      Line 120 “G. lucidum FP-58537-Sp and P. velutina” should be written in italics. And P. velutina does not have a strain number.

2.      Lines 274-275 – “…phosphate buffer consisting of KH2PO4 5.3 g/L and K2HPO4 10.6 g/L in an ultra-pure water solution.” – the pH of the buffer must be specified.

3.      Line 289 - 16SrDNA – is missing a space.

4.  About Supplementary materials:

      4.1. Supplementary materials- “Table S2. TCEP residual concentrations and laccase activities detected in the liquid phase” – for some values there is no standard deviation.

    4.2. Supplementary materials - “Table S3. Chromatographic characteristics of the TPs of TBP by T. versicolor and G. lucidum in Erlenmeyer flasks after 7d incubation”,

“Table S4. Chromatographic characteristics of the TPs of TCEP by T. versicolor and G. lucidum”, “Table S5. COD and turbidity at the end of each 3d batch treatment in the TBR” – there is no standard deviation.

    4.3. “Table S4. Chromatographic characteristics of the TPs of TCEP by T. versicolor and G. lucidum” - Monoisotopic mass – no units of measurement.

    4.4. Supplementary materials  - “Figure S1 The Fungal DGGE profiles” on page 2 and “Figure S1. DGGE profiles of the fungal (A) and bacterial (B) communities found in the different treatments, each square mark a band that was recovered, sequenced, and uploaded to NCBI, colour indicates the taxonomical group at genus level of the closest match for that sequence. CW: colonized wood, KC: killed control, AC: abiotic control, EX: experimental treatment.” – page 7 - This is a duplication.

     4.5. Supplementary materials   - “Figure S2 The dendrograms generated from the hierarchical clustering analysis of the DGGE banding patterns” on page 3 duplicates “Figure S2. Dendrograms for fungal (A) and bacterial (B) communities of the different treatments, clustered with UPGMA from the DGGE similarity matrix calculated with Dice index. CW: colonized wood, KC: killed control, C: abiotic control, EX: experimental treatment.” on page 8.

      4.6. Supplementary materials    - “Table S8. Standard curve results for the qPCRs performed in this study” – there is no standard deviation in the values.

     4.7. Supplementary materials    - “Table S8. Standard curve results for the qPCRs performed in this study” - values cannot be more than 100 ‘Efficiency’ equal to 101.27% and 100.03% is not correct data.

5.      Line 308 – “…for 16h at 60°C and 16h.” – Repetition is present.

6.      Line 320 – “Quantitative PCR of total bacteria, total fungi, and specifically for Ganoderma sp….” and Lines 326-327 – “…and reverse primers for total bacteria, total fungi, and Ganoderma sp., respectively, and 36 ng DNA for most samples.” – what kind of bacteria do you mean?

7.      Line 357 – UNIGAM 8625 – the country of the manufacturer must be added.

8.      Figure 1 - The caption of the ordinate axis needs to be clarified or a transcript needs to be added to the figure caption.

9.      Section “3.1.3. Transformation products generated during the TBP and TCEP degradation” - Do the authors have a diagram of the degradation pathway of TBP and TCEP by the strains studied? It would be logical to include this information both in the introduction and in the results if new data on metabolites are obtained.

10.  Figure 2 - no standard deviations.

11.  Table 3 has no standard deviations.

12.  Lines 497-515 - all bacterial names should be written in italics.

13.  It is not clear why the authors decided to analyse the composition of the bacterial communities of wood chips.

14.  What is the novelty of your research compared to a published article [Losantos D, Sarra M, Caminal G. OPFR removal by white rot fungi: screening of removers and approach to the removal mechanism. Front Fungal Biol. 2024 May 17;5:1387541. doi: 10.3389/ffunb.2024.1387541. PMID: 38827887; PMCID: PMC11140845]

Why didn't you list it as a backstory (no link to it)?

15.  Line 264 - under ‘Materials and methods’ there is a subsection “2.8.5. Toxicity assessment”, However, this information is not available in the ‘Results’ section

16.   There is no ‘Discussion’ section in the article.

Author Response

We appreciate the effort in reviewing our manuscript.

Find enclosed responses to your questions and comments in the atacked file.

Thank you for an interesting article, but in my opinion, it needs significant revision. As I read it, I have a number of questions and comments.

Thank you very much for taking the time to review this manuscript. Please find the detailed responses below. The manuscript has been changed significatively because we did not separate the results section and the discussion section. As consequently the changes marked were too much to the readable. So, we include the revised version in the clean form but if reviewer prefer with changes marked version, we can provide it.

Thank you for the comments. We tried to do our best to answer your questions.

  1. Line 120 “G. lucidum FP-58537-Sp and P. velutina” should be written in italics. And  velutinadoes not have a strain number.

We apologise for the formal mistake and we correct the origin of our P. velutina strain and the number strain has been added.

T. versicolor ATCC 42530 was acquired from the American Type Culture collection, P. velutina FBCC941 (T244i) was obtained from the Fungal Biotechnology Culture Collection of the University of Helsinki (FBCC, Helsinki, Finland) and G. lucidum FP-58537-Sp was kindly provided by Dr. C.A. Reddy from the United States Department of Agriculture Forest Products Laboratory”

  1. Lines 274-275 – “…phosphate buffer consisting of KH2PO4 5.3 g/L and K2HPO4 10.6 g/L in an ultra-pure water solution.” – the pH of the buffer must be specified.

Thank you for drawing our attention to this, the buffer pH of 7 has been specified in the text.

  1. Line 289 - 16SrDNA – is missing a space.

We have added the missing space following the reviewer's suggestions.

  1. About Supplementary materials:

      4.1. Supplementary materials- “Table S2. TCEP residual concentrations and laccase activities detected in the liquid phase” – for some values there is no standard deviation.

The standard deviation has been calculated for the removal percentage. For laccase, we do not have enough values to calculate standard deviation because after 7 days the laccase is almost deactivated, consequently the activity is low or no detectable.

    4.2. Supplementary materials - “Table S3. Chromatographic characteristics of the TPs of TBP by T. versicolor and G. lucidum in Erlenmeyer flasks after 7d incubation”,

“Table S4. Chromatographic characteristics of the TPs of TCEP by T. versicolor and G. lucidum”,

“Table S5. COD and turbidity at the end of each 3d batch treatment in the TBR” – there is no standard deviation.

At the end of each batch, only one sample was taken because the collecting tank was perfectly agitated and consequently it was homogenous. So, we cannot calculate the standard deviation of the measurement but the most relevant is the evolution along the repeated batch treatment, this is the end of 10 cycles.

    4.3. “Table S4. Chromatographic characteristics of the TPs of TCEP by T. versicolor and G. lucidum” - Monoisotopic mass – no units of measurement.

Thank you for the comment. Units have been added.

    4.4. Supplementary materials  - “Figure S1 The Fungal DGGE profiles” on page 2 and “Figure S1. DGGE profiles of the fungal (A) and bacterial (B) communities found in the different treatments, each square mark a band that was recovered, sequenced, and uploaded to NCBI, colour indicates the taxonomical group at genus level of the closest match for that sequence. CW: colonized wood, KC: killed control, AC: abiotic control, EX: experimental treatment.” – page 7 - This is a duplication.

This formatting error has been corrected and duplications in Figure S2 from supplementary materials have been removed.

     4.5. Supplementary materials   - “Figure S2 The dendrograms generated from the hierarchical clustering analysis of the DGGE banding patterns” on page 3 duplicates “Figure S2. Dendrograms for fungal (A) and bacterial (B) communities of the different treatments, clustered with UPGMA from the DGGE similarity matrix calculated with Dice index. CW: colonized wood, KC: killed control, C: abiotic control, EX: experimental treatment.” on page 8.

This formatting error has been corrected and duplications in Figure S3 from supplementary materials have been removed.

      4.6. Supplementary materials    - “Table S8. Standard curve results for the qPCRs performed in this study” – there is no standard deviation in the values.

We followed the guidelines for the validation of the qPCR results for the three primer pairs used. First, we optimized the methodology for each primer set and performed several tests to corroborate the efficiency of the qPCRs and the linearity of the standard curve. Regarding the results shown in Table 8, these correspond to the final analysis performed on all samples, in triplicate. In addition, three technical replicates were carried out for each replicate, so that for the purposes of analysis 9 replicates were used. In this case, all these replicates were analyzed in a single qPCR with a single standard curve. We consider more appropriate to show the information directly linked to the data presented, so that for this assay the efficiency and the linearity of the standard curve do not present standard deviation.

     4.7. Supplementary materials    - “Table S8. Standard curve results for the qPCRs performed in this study” - values cannot be more than 100 ‘Efficiency’ equal to 101.27% and 100.03% is not correct data.

The efficiency of qPCR depends on the assay, the performance of the master mix and the quality of the sample. It is calculated based on a standard curve where the slope of the regression line is related to the amplification efficiency. If it is 100%, it means that the amount of PCR reaction product is doubling with each cycle, but this is not always the case; so according to good quality standards, efficiency is generally considered to be between 90% and 110%, as reported in the literature [1]. In the assays performed, both the efficiency is close to the theoretical 100% and linearity equal to or greater than 99%, which reinforces the validity of our results

  1. Line 308 – “…for 16h at 60°C and 16h.” – Repetition is present.

We have corrected the typo following the reviewer's suggestions.

  1. Line 320 – “Quantitative PCR of total bacteria, total fungi, and specifically for Ganoderma sp….” and Lines 326-327 – “…and reverse primers for total bacteria, total fungi, and Ganoderma sp., respectively, and 36 ng DNA for most samples.” – what kind of bacteria do you mean?

The term total bacteria refer to all bacterial populations amplified with the universal primer pair for the Bacteria domain used. Considering that indigenous bacterial populations can have an antagonistic effect on Ganoderma, it is essential to quantify them globally by qPCR and compare them with the total fungal populations and, within the latter, with those belonging to Ganoderma. We appreciate your comments and a clearer explanation has been included in the text.

  1. Line 357 – UNIGAM 8625 – the country of the manufacturer must be added.

(Manchester, United Kingdom) has been added.

  1. Figure 1 - The caption of the ordinate axis needs to be clarified or a transcript needs to be added to the figure caption.

Transcript has been added to the figure caption.

  1. Section “3.1.3. Transformation products generated during the TBP and TCEP degradation” - Do the authors have a diagram of the degradation pathway of TBP and TCEP by the strains studied? It would be logical to include this information both in the introduction and in the results if new data on metabolites are obtained.

 This aspect has been included in the discussion section as well as the corresponding reference.

  1. Figure 2 - no standard deviations.
  2. Table 3 has no standard deviations.

At the end of each batch, only one sample was taken because the collecting tank was perfectly agitated and consequently it was homogenous. So, we cannot calculate the standard deviation of the measurement but the most relevant is the evolution along the repeated batch treatment, this is the end of 10 cycles.

  1. Lines 497-515 - all bacterial names should be written in italics.

Names above the genus level have traditionally been written in Roman type, without italics. We are sorry, but we were not aware that this journal had adopted the practice of using italics for all taxonomic ranks. Therefore, bacterial nomenclature has been revised and the format corrected to italics following the reviewers’ suggestions.

  1. It is not clear why the authors decided to analyse the composition of the bacterial communities of wood chips.

This research aims to assess the capacity of Ganoderma lucidum to degrade OPFRs using a real wastewater where different indigenous bacterial and fungal populations can interact with the fungus immobilized in the oak wood chips of the bioreactors.

Taking into account that these bioreactors operated in non-sterile conditions with real wastewater, it was considered important to characterize the native microbial populations of the influent wastewater, since they could also colonize the lignocellulosic support, and compete with Ganoderma for the same ecological niche. For this reason, in addition to assessing the persistence of Ganoderma by qPCR, it was a priority to characterize the bacterial populations that had colonized the substrate. Furthermore, since the abiotic and killed controls, where Ganoderma had not been immobilized or was dead, respectively, proved to be very efficient in removing the studied contaminants, it was also considered crucial to know which bacterial populations could be potentially responsible for the OPFRs degradation.

  1. What is the novelty of your research compared to a published article [Losantos D, Sarra M, Caminal G. OPFR removal by white rot fungi: screening of removers and approach to the removal mechanism. Front Fungal Biol. 2024 May 17;5:1387541. doi: 10.3389/ffunb.2024.1387541. PMID: 38827887; PMCID: PMC11140845]

Why didn't you list it as a backstory (no link to it)?

The comment is pertinent because both studies were carried out in parallel and related. So the main findings of this paper have been added to improve the discussion section.

  1. Line 264 - under ‘Materials and methods’ there is a subsection “2.8.5. Toxicity assessment”, However, this information is not available in the ‘Results’ section

The toxicity results are provided in lines 481-483.

  1. There is no ‘Discussion’ section in the article.

We apologise for the formal mistake of non-presenting the results and discussion in separated sections. In the revised version, we hope that the discussion is clearer.

Reference

  1. Broeders, S.; Huber, I.; Grohmann, L.; Berben, G.; Taverniers, I.; Mazzara, M.; Roosens, N.; Morisset, D. Guidelines for Validation of Qualitative Real-Time PCR Methods. Trends Food Sci Technol 2014, 37, 115–126, doi:10.1016/j.tifs.2014.03.008.

Round 2

Reviewer 1 Report

The manuscript was sufficiently improved.

See major comments.

Author Response

We are happy to know that all concerts about the manuscript have been adressed.

All of authors appreviate the help for improving the manuscript.

Reviewer 2 Report

Presented article is an important contribution to the scientific discussion on this topic. The research is appropriately designed and the methods are described in detail. The results in the main article are well described and easy to interpret. The authors included supplementary material to enable evaluation and interpretation of the study results. Unfortunately, the lack of attached toxicity results and minor editorial errors cause the article to be corrected before publication in JoF. Therefore please attach detailed toxicity analysis results, in particular EC50 values.

Editorial errors:

-            Table S5 should be formatted similarly to the other tables

-            Table 2 should be moved so that it is not divided into two pages

-            L135: no italic in Latin name of T. versicolor as well as in References: 18, 23, 25,33,34,35,36,37,48,50,51,53,57,63,66,73 etc.

-            L329 and others in the remaining text: remove letter A from laccase activity unit, should be U/L instead of AU/L

-            L381: should be mM/cm or mM cm-1

-            remove italics from the sentence ”on oak wood chips”

-            L448: bis not Bis

Author Response

We apologise for the editing mistakes and we appreciate your deep review which allowed us to improve our manuscript.

Find enclosed the answer to your comments in the attached file.

Related to the tittle, we prefer to highlight the fungus Ganoderma as OPFRs degrader, because we select it for the application on the bioreactor.

Reviewer 3 Report

Dear Authors,

Thank you for making a number of changes to your article, however I still have a few minor comments.

1.    Line 118 – the name of the microorganism ‘P. velutina...’ should be written in italics.

     Line 135 – the name of the microorganism ‘T. versicolor’  - should be written in italics.

2.    Line 257 –“20ºC” - is missing a space.

3.    Line 312 – “1,5 mM” replace with “1.5 mM”.

4.    Line 340 “…immobilized on oak wood chips…” – not in italics.

5.    Figure 1 - It would be better to replace the values on the Y-axis in 0.2 format, i.e. with a dot rather than a comma, in accordance with the JoF rules.

6.    Line 578 – 24,1%” replace with “24.1%”.

7.    Supplementary material - all table and figure names should have a full stop at the end.

Author Response

We apologise for the editing mistakes and we appreciate your deep review which allowed us to improve our manuscript.

Find enclosed the answer to your comments in the attached file.
